# Machine learning detects terminal singularities

**Tom Coates**
Department of Mathematics
Imperial College London
180 Queen's Gate
London, SW7 2AZ
UK
t.coates@imperial.ac.uk

**Alexander M. Kasprzyk**
School of Mathematical Sciences
University of Nottingham
Nottingham, NG7 2RD
UK
a.m.kasprzyk@nottingham.ac.uk

**Sara Veneziale**[*]
Department of Mathematics
Imperial College London
180 Queen's Gate
London, SW7 2AZ
UK
s.veneziale21@imperial.ac.uk

## Abstract

Algebraic varieties are the geometric shapes defined by systems of polynomial equations; they are ubiquitous across mathematics and science. Amongst these algebraic varieties are $\mathbb{Q}$-Fano varieties: positively curved shapes which have $\mathbb{Q}$-factorial terminal singularities. $\mathbb{Q}$-Fano varieties are of fundamental importance in geometry as they are 'atomic pieces' of more complex shapes – the process of breaking a shape into simpler pieces in this sense is called the Minimal Model Programme.

Despite their importance, the classification of $\mathbb{Q}$-Fano varieties remains unknown. In this paper we demonstrate that machine learning can be used to understand this classification. We focus on eight-dimensional positively-curved algebraic varieties that have toric symmetry and Picard rank two, and develop a neural network classifier that predicts with 95% accuracy whether or not such an algebraic variety is $\mathbb{Q}$-Fano. We use this to give a first sketch of the landscape of $\mathbb{Q}$-Fano varieties in dimension eight.

How the neural network is able to detect $\mathbb{Q}$-Fano varieties with such accuracy remains mysterious, and hints at some deep mathematical theory waiting to be uncovered. Furthermore, when visualised using the quantum period, an invariant that has played an important role in recent theoretical developments, we observe that the classification as revealed by ML appears to fall within a bounded region, and is stratified by the Fano index. This suggests that it may be possible to state and prove conjectures on completeness in the future.

Inspired by the ML analysis, we formulate and prove a new global combinatorial criterion for a positively curved toric variety of Picard rank two to have terminal singularities. Together with the first sketch of the landscape of $\mathbb{Q}$-Fano varieties in higher dimensions, this gives strong new evidence that machine learning can be an essential tool in developing mathematical conjectures and accelerating theoretical discovery.

---

[*]Corresponding author.

# 1  Introduction

Systems of polynomial equations occur throughout mathematics and science; see e.g. [3, 17, 18, 35]. Solutions of these systems define shapes called *algebraic varieties*. Depending on the equations involved, algebraic varieties can be smooth (as in Figure 1a) or have singularities (as in Figures 1b and 1c). In this paper we show that machine learning methods can detect a class of singularities called *terminal singularities*.

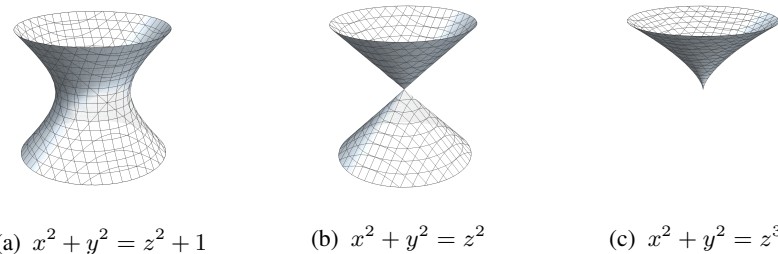

(a) $x^2 + y^2 = z^2 + 1$        (b) $x^2 + y^2 = z^2$        (c) $x^2 + y^2 = z^3$

Figure 1: Algebraic varieties in $\mathbb{R}^3$ with different defining equations.

A key class of algebraic varieties are *Fano varieties*: positively curved shapes that are basic building blocks in algebraic geometry. Fano varieties are 'atomic pieces' of more complex shapes, in the sense of the Minimal Model Programme [7, 25, 26]. Running the Minimal Model Programme – that is, breaking an algebraic variety $X$ into atomic pieces – involves making birational transformations of $X$. These are modifications on subsets with zero volume (and codimension at least one), and can either introduce or remove singularities. The building blocks that emerge from this process are not necessarily smooth: they satisfy a weaker condition called $\mathbb{Q}$-factoriality,[2] and can have mild singularities called terminal singularities [38]. Fano varieties that are $\mathbb{Q}$-factorial and have terminal singularities are called $\mathbb{Q}$-*Fano varieties*.

The classification of $\mathbb{Q}$-Fano varieties is therefore a long-standing problem of great importance [4, 15, 27, 33, 34] – one can think of this as building a Periodic Table for geometry. But, despite more than a century of study, very little is known. In what follows we exploit the fact that machine learning can detect terminal singularities to give the first sketch of part of the classification of higher-dimensional $\mathbb{Q}$-Fano varieties.

We probe the classification of $\mathbb{Q}$-Fano varieties using a class of highly-symmetrical shapes called *toric varieties*. (For example, the algebraic varieties pictured in Figure 1 are toric varieties.) Toric varieties are particularly suitable for computation and machine learning, because their geometric properties are encoded by simple combinatorial objects. We consider Fano toric varieties of Picard rank two. These can be encoded using a $2 \times N$ matrix of non-negative integers called the *weight matrix*; here the dimension of the toric variety is $N - 2$.

To determine whether such a toric variety $X$ is a $\mathbb{Q}$-Fano variety we need to check whether $X$ is $\mathbb{Q}$-factorial, and whether the singularities of $X$ are terminal. Checking $\mathbb{Q}$-factoriality from the weight matrix of $X$ turns out to be straightforward (see §3) but checking terminality is extremely challenging. This is because there is no satisfactory theoretical understanding of the problem. We lack a global criterion for detecting terminality in terms of weight data (such as [24] in a simpler setting) and so have to fall back on first enumerating all the singularities to analyse, and then checking terminality for each singularity. Each step is a challenging problem in discrete geometry: the first step involves building a different combinatorial object associated to the $n$-dimensional toric variety $X$, which is a collection of cones in $\mathbb{R}^n$ called the *fan* $\Sigma(X)$; the second step involves checking for various cones in the fan whether or not they contain lattice points on or below a certain hyperplane. To give a sense of the difficulty of the computations involved, generating and post-processing our dataset of 10 million toric varieties in dimension eight took around 30 CPU years.

---

[2]An algebraic variety $X$ is $\mathbb{Q}$-factorial if it is normal and, in addition, for each rank-one reflexive sheaf $E$ on $X$, some tensor power of $E$ is a line bundle. This implies that the dimension of the singular locus in $X$ is at most $\dim X - 2$, and that some tensor power of the canonical sheaf (of top-degree differential forms) is a line bundle.

To overcome this difficulty, and hence to begin to investigate the classification of $\mathbb{Q}$-Fano varieties in dimension eight, we used supervised machine learning. We trained a feed-forward neural network classifier on a balanced dataset of 5 million examples; these are eight-dimensional $\mathbb{Q}$-factorial Fano toric varieties of Picard rank two, of which 2.5 million are terminal and 2.5 million non-terminal. Testing on a further balanced dataset of 5 million examples showed that the neural network classifies such toric varieties as terminal or non-terminal with an accuracy of 95%. This high accuracy allowed us to rapidly generate many additional examples that are with high probability $\mathbb{Q}$-Fano varieties – that is, examples that the classifier predicts have terminal singularities. This ML-assisted generation step is much more efficient: generating 100 million examples in dimension eight took less than 120 CPU hours.

The fact that the ML classifier can detect terminal singularities with such high accuracy suggests that there is new mathematics waiting to be discovered here – there should be a simple criterion in terms of the weight matrix to determine whether or not a toric variety $X$ has terminal singularities. In §5 we take the first steps in this direction, giving in Algorithm 1 a new method to check terminality directly from the weight matrix, for toric varieties of Picard rank two. A proof of correctness is given in the Supplementary Material. This new algorithm is fifteen times faster than the naïve approach that we used to generate our labelled dataset, but still several orders of magnitude slower than the neural network classifier. We believe that this is not the end of the story, and that the ML results suggest that a simpler criterion exists. Note that the neural network classifier cannot be doing anything analogous to Algorithm 1: the algorithm relies on divisibility relations between entries of the weight matrix (GCDs etc.) that are not visible to the neural network, as they are destroyed by the rescaling and standardisation that is applied to the weights before they are fed to the classifier.

In §6 we use the ML-assisted dataset of 100 million examples to begin to explore the classification of $\mathbb{Q}$-Fano varieties in dimension eight. We visualise the dataset using the *regularized quantum period*, an invariant that has played an important role in recent theoretical work on $\mathbb{Q}$-Fano classification, discovering that an appropriate projection of the data appears to fill out a wedge-shaped region bounded by two straight lines. This visualisation suggests some simple patterns in the classification: for example, the distance from one edge of the wedge appears to be determined by the Fano index of the variety.

Our work is further evidence that machine learning can be an indispensable tool for generating and guiding mathematical understanding. The neural network classifier led directly to Algorithm 1, a new theoretical result, by revealing that the classification problem was tractable and thus there was probably new mathematics waiting to be found. This is part of a new wave of application of artificial intelligence to pure mathematics [10, 14, 16, 20, 39–41], where machine learning methods drive theorem discovery.

A genuinely novel contribution here, though, is the use of machine learning for data generation and data exploration in pure mathematics. Sketching the landscape of higher-dimensional $\mathbb{Q}$-Fano varieties using traditional methods would be impossible with the current theoretical understanding, and prohibitively expensive using the current exact algorithms. Training a neural network classifier however, allows us to explore this landscape easily – a landscape that is unreachable with current mathematical tools.

**Why dimension eight?** We chose to work with eight-dimensional varieties for several reasons. It is important to distance ourselves from the surface case (dimension two), where terminality is a trivial condition. A two-dimensional algebraic variety has terminal singularities if and only if it is smooth. On the other hand, we should consider a dimension where we can generate a sufficient amount of data for machine learning (the analogue of our dataset in dimension three, for example, contains only 34 examples [23]) and where we can generate enough data to meaningfully probe the classification. Moreover, we work in Picard rank two because there already exists a fast combinatorial formula to check terminality in rank one [24]; Picard rank two is the next natural case to consider.

**Data and code availability** The datasets underlying this work and the code used to generate them are available from Zenodo under a CC0 license [11]. Data generation and post-processing was carried out using the computational algebra system Magma V2.27-3 [5]. The machine learning model was built using PyTorch v1.13.1 [36] and scikit-learn v1.1.3 [37]. All code used and trained models are available from BitBucket under an MIT licence [12].

## 2 Mathematical background

The prototypical example of a Fano variety is projective space $\mathbb{P}^{N-1}$, which can be thought of as the quotient of $\mathbb{C}^N \setminus \{\mathbf{0}\}$ by $\mathbb{C}^\times$ acting as follows:

$$\lambda \cdot (z_1, \ldots, z_N) = (\lambda z_1, \ldots, \lambda z_N)$$

Fano toric varieties of Picard rank two arise similarly. They can be constructed as the quotient of $\mathbb{C}^N \setminus S$, where $S$ is a union of subspaces, by an action of $(\mathbb{C}^\times)^2$. This action, and the union of subspaces $S$, is encoded by a weight matrix:

$$\begin{bmatrix} a_1 & \cdots & a_N \\ b_1 & \cdots & b_N \end{bmatrix} \tag{2.1}$$

Here we assume that all $(a_i, b_i) \in \mathbb{Z}^2 \setminus \{\mathbf{0}\}$ lie in a strictly convex cone $C \subset \mathbb{R}^2$. The action is

$$(\lambda, \mu) \cdot (z_1, \ldots, z_N) = (\lambda^{a_1} \mu^{b_1} z_1, \ldots, \lambda^{a_N} \mu^{b_N} z_N)$$

and $S = S_+ \cup S_-$ is the union of subspaces $S_+$ and $S_-$, where

$$\begin{aligned} S_+ &= \{(z_1, \ldots, z_N) \mid z_i = 0 \text{ if } b_i/a_i > b/a\} \\ S_- &= \{(z_1, \ldots, z_N) \mid z_i = 0 \text{ if } b_i/a_i < b/a\} \end{aligned} \tag{2.2}$$

and $a = \sum_{i=1}^N a_i$, $b = \sum_{i=1}^N b_i$: see [6]. The quotient $X = (\mathbb{C}^N \setminus S)/(\mathbb{C}^\times)^2$ is an algebraic variety of dimension $N - 2$. We assume in addition that both $S_+$ and $S_-$ have dimension at least two; this implies that the second Betti number of $X$ is two, that is, $X$ has Picard rank two.

Since we have insisted that all columns $(a_i, b_i)$ lie in a strictly convex cone $C$, we can always permute columns and apply an $\mathrm{SL}_2(\mathbb{Z})$ transformation to the weight matrix to obtain a matrix in standard form:

$$\begin{bmatrix} a_1 & a_2 & \cdots & a_N \\ 0 & b_2 & \cdots & b_N \end{bmatrix} \tag{2.3}$$

where all entries are non-negative, the columns are cyclically ordered anticlockwise, and $a_N < b_N$. This transformation corresponds to renumbering the co-ordinates of $\mathbb{C}^N$ and reparametrising the torus $(\mathbb{C}^\times)^2$ that acts, and consequently leaves the quotient variety $X$ that we construct unchanged.

We will consider weight matrices (2.1) that satisfy an additional condition called being *well-formed*. An $r \times N$ weight matrix is called standard if the greatest common divisor of its $r \times r$ minors is one, and is well-formed if every submatrix formed by deleting a column is standard [2]. Considering only well-formed weight matrices guarantees that a toric variety determines and is determined by its weight matrix, uniquely up to $\mathrm{SL}_r(\mathbb{Z})$-transformation.

**Testing terminality**  As mentioned in the introduction, an $n$-dimensional toric variety $X$ determines a collection $\Sigma(X)$ of cones in $\mathbb{R}^n$ called the fan of $X$. A toric variety is completely determined by its fan. The process of determining the fan $\Sigma(X)$ from the weight matrix (2.1) is explained in the Supplementary Material; this is a challenging combinatorial calculation. In the fan $\Sigma(X)$, the one-dimensional cones are called rays. For a Fano toric variety $X$, taking the convex hull of the first lattice point on each ray defines a convex polytope $P$, and $X$ has terminal singularities if and only if the only lattice points in $P$ are the origin and the vertices. Verifying this is a conceptually straightforward but computationally challenging calculation in integer linear programming.

## 3 Data generation

We generated a balanced, labelled dataset of ten million $\mathbb{Q}$-factorial Fano toric varieties of Picard rank two and dimension eight. These varieties are encoded, as described above, by weight matrices. We generated $2 \times 10$ integer-valued matrices in standard form, as in (2.3), with entries chosen uniformly at random from the set $\{0, \ldots, 7\}$. Minor exceptions to this were the values for $a_1$ and $b_N$, which were both chosen uniformly at random from the set $\{1, \ldots, 7\}$, and the value for $a_N$, which was chosen uniformly at random from the set $\{0, \ldots, b_N - 1\}$. Once a random weight matrix was generated, we retained it only if it satisfied:

Table 1: Final network architecture and configuration.

| Hyperparameter | Value | Hyperparameter | Value |
|---:|---|---:|---|
| Layers | $(512, 768, 512)$ | Momentum | 0.99 |
| Batch size | 128 | LeakyRelu slope | 0.01 |
| Initial learning rate | 0.01 | | |

1. None of the columns are the zero vector.

2. The sum of the columns is not a multiple of any of them.

3. The subspaces $S_+$ and $S_-$ in (2.2) are both of dimension at least two.

4. The matrix is well-formed.

The first condition here was part of our definition of weight matrix; the second condition is equivalent to $X$ being $\mathbb{Q}$-factorial; the third condition guarantees that $X$ has Picard rank two; and the fourth condition was discussed above.

We used rejection sampling to ensure that the dataset contains an equal number of terminal and non-terminal examples. Before generating any weight matrix, a boolean value was set to `True` (terminal) or `False` (non-terminal). Once a random weight matrix that satisfied conditions (1)–(4) above was generated, we checked if the corresponding toric variety was terminal using the method discussed in §2. If the terminality check agreed with the chosen boolean, the weight matrix was added to our dataset; otherwise the generation step was repeated until a match was found.

As discussed, different weight matrices can give rise to the same toric variety. Up to isomorphism, however, a toric variety $X$ is determined by the isomorphism class of its fan. We deduplicated our dataset by placing the corresponding fan $\Sigma(X)$, which we had already computed in order to test for terminality, in normal form [19, 29]. In practice, very few duplicates occurred.

## 4 Building the machine learning model

We built a neural network classifier to determine whether a $\mathbb{Q}$-factorial Fano variety of Picard rank two and dimension eight is terminal. The network was trained on the features given by concatenating the two rows of a weight matrix, $[a_1, \ldots, a_{10}, b_1, \ldots, b_{10}]$. The features were standardised by translating their mean to zero and scaling to variance one. The network, a multilayer perceptron, is a fully connected feedforward neural network with three hidden layers and leaky ReLu activation function. It was trained on the dataset described in §3 using binary cross-entropy as loss function, stochastic mini-batch gradient descent optimiser and using early-stopping, for a maximum of 150 epochs and with learning rate reduction on plateaux. We tested the model on a balanced subset of 50% of the data (5M); the remainder was used for training (40%; 4M balanced) and validation (10%; 1M).

Hyperparameter tuning was partly carried out using RayTune [31] on a small portion of the training data, via random grid search with Async Successive Halving Algorithm (ASHA) scheduler [30], for 100 experiments. Given the best configuration resulting from the random grid search, we then manually explored nearby configurations and took the best performing one. The final best network configuration is summarised in Table 1.

By trying different train-test splits, and using 20% of the training data for validation throughout, we obtained the learning curve in Figure 2a. This shows that a train-validate-test split of 4M-1M-5M produced an accurate model that did not overfit. Training this model gave the loss learning curve in Figure 2b, and a final accuracy (on the test split of size 5M) of 95%.

## 5 Theoretical result

The high accuracy of the model in §4 was very surprising. As explained in the introduction, $\mathbb{Q}$-Fano varieties are of fundamental importance in algebraic geometry. However, asking whether a Fano variety has terminal singularities is, in general, an extremely challenging geometric question. In the case of a Fano toric variety one would typically proceed by constructing the fan, and then performing

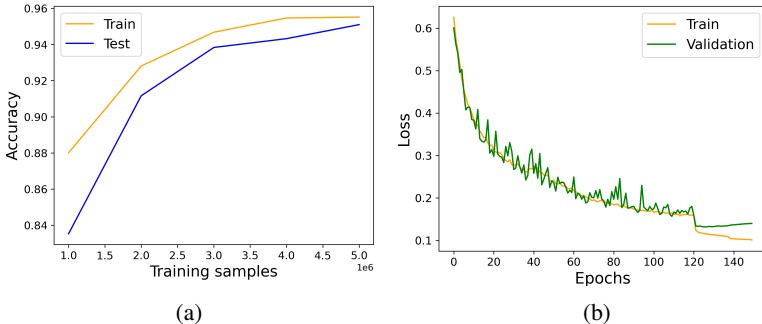

Figure 2: (a) Accuracy for different train-test splits; (b) epochs against loss for the network trained on 5M samples.

a cone-by-cone analysis of the combinatorics. This is computationally expensive and unsatisfying from a theoretical viewpoint. The success of the model suggested that a more direct characterisation is possible from the weight matrix alone. An analogous characterisation exists in the simpler case of weighted projective spaces [24], which have Picard rank one, however no such result in higher Picard rank was known prior to training this model.

Inspired by this we prove a theoretical result, Proposition 5.3, which leads to a new algorithm for checking terminality directly from the weight matrix, for $\mathbb{Q}$-factorial Fano toric varieties of Picard rank two. Consider a weight matrix as in (2.1) that satisfies conditions (1)–(4) from §3, and the toric variety $X$ that it determines. As discussed in §2, and explained in detail in the Supplementary Material, $X$ determines a convex polytope $P$ in $\mathbb{R}^{N-2}$, with $N$ vertices given by the first lattice points on the $N$ rays of the fan. Each of the vertices of $P$ is a lattice point (i.e., lies in $\mathbb{Z}^{N-2} \subset \mathbb{R}^{N-2}$), and $X$ has terminal singularities if and only if the only lattice points in $P$ are the vertices $e_1, \ldots, e_N$ and the origin.

**Definition 5.1.** Let $\Delta_i$ denote the simplex in $\mathbb{R}^{N-2}$ with vertices $e_1, \ldots, \hat{e}_i, \ldots, e_N$ where $e_i$ is omitted. We say that $\Delta_i$ is *mostly empty* if each lattice point in $\Delta_i$ is either a vertex or the origin.

**Notation 5.2.** Let $\{x\}$ denote the fractional part $x - \lfloor x \rfloor$ of a rational number $x$.

**Proposition 5.3.** *Consider a weight matrix*

$$\begin{bmatrix} a_1 & \cdots & a_N \\ b_1 & \cdots & b_N \end{bmatrix}$$

*that satisfies conditions 1–4 from §3. Let $g_i = \gcd\{a_i, b_i\}$, and let $A_i$, $B_i$ be integers such that $A_i a_i + B_i b_i = g_i$. Set*

$$\alpha_i^j = \frac{a_j b_i - b_j a_i}{g_i} \qquad\qquad \alpha_i = \sum_{j=1}^N \alpha_i^j$$

$$\beta_i^j = -A_i a_j - B_i b_j \qquad\qquad \beta_i = \sum_{j=1}^N \beta_i^j \qquad\qquad f_i = \frac{\alpha_i g_i}{\gcd\{g_i, \beta_i\}}$$

*noting that all these quantities are integers. Then $\Delta_i$ is mostly empty if and only if for all $k \in \{0, \ldots, f_i - 1\}$ and $l \in \{0, \ldots, g_i - 1\}$ such that*

$$\sum_{j=1}^N \left\{ k \frac{\alpha_i^j}{f_i} + l \frac{\beta_i^j}{g_i} \right\} = 1$$

*we have that*

$$\left\{ k \frac{\alpha_i^j}{f_i} + l \frac{\beta_i^j}{g_i} \right\} = \left\{ \frac{\alpha_i^j}{\alpha_i} \right\}$$

*for all $j$.*

Let $s_+ = \{i \mid a_i b - b_i a > 0\}$, $s_- = \{i \mid a_i b - b_i a < 0\}$, and let $I$ be either $s_+$ or $s_-$. Then $\Delta_i$, $i \in I$, forms a triangulation of $P$. Thus $X$ has terminal singularities if and only if $\Delta_i$ is mostly empty for each $i \in I$. This leads to Algorithm 1.

---

**Algorithm 1** Test terminality for weight matrix $W = [[a_1, \ldots, a_N], [b_1, \ldots, b_N]]$.

---

1: Set $a = \sum_{i=1}^N a_i$, $b = \sum_{i=1}^N b_i$.
2: Set $s_+ = \{i \mid a_i b - b_i a > 0\}$ and $s_- = \{i \mid a_i b - b_i a < 0\}$.
3: Set $I$ to be the smaller of $s_+$ and $s_-$.
4: **for** $i \in I$ **do**
5:     Test if $\Delta_i$ is mostly empty, using Proposition 5.3.
6:     **if** $\Delta_i$ is not mostly empty **then**
7:         return False.
8:     **end if**
9: **end for**
10: return True.

---

**Comparisons**  Testing on 100 000 randomly-chosen examples indicates that Algorithm 1 is approximately 15 times faster than the fan-based approach to checking terminality that we used when labelling our dataset (0.020s per weight matrix for Algorithm 1 versus 0.305s for the standard approach implemented in Magma). On single examples, the neural network classifier is approximately 30 times faster than Algorithm 1. The neural network also benefits greatly from batching, whereas the other two algorithms do not: for batches of size 10 000, the neural network is roughly 2000 times faster than Algorithm 1.

## 6  The terminal toric Fano landscape

Having trained the terminality classifier, we used it to explore the landscape of $\mathbb{Q}$-Fano toric varieties with Picard rank two. To do so, we built a large dataset of examples and analysed their *regularized quantum period*, a numerical invariant of $\mathbb{Q}$-Fano varieties [8]. For smooth low-dimensional Fano varieties, it is known that the regularized quantum period is a complete invariant [9]. This is believed to be true in higher dimension, but is still conjectural. Given a $\mathbb{Q}$-Fano variety $X$, its regularized quantum period is a power series

$$\hat{G}_X(t) = \sum_{d=0}^\infty c_d t^d$$

where $c_0 = 1$, $c_1 = 0$, $c_d = d!\, r_d$, and $r_d$ is the number of degree-$d$ rational curves in $X$ that satisfy certain geometric conditions. Formally speaking, $r_d$ is a degree-$d$, genus-zero Gromov–Witten invariant [28]. The *period sequence* of $X$ is the sequence $(c_d)$ of coefficients of the regularized quantum period. This sequence grows rapidly. In the case where $X$ is a $\mathbb{Q}$-Fano toric variety of Picard rank two, rigorous asymptotics for this growth are known.

**Theorem 6.1** (Theorem 5.2, [10]). *Consider a weight matrix*

$$\begin{bmatrix} a_1 & \ldots & a_N \\ b_1 & \ldots & b_N \end{bmatrix}$$

*for a $\mathbb{Q}$-factorial Fano toric variety $X$ of Picard rank two. Let $a = \sum_{i=1}^N a_i$ and $b = \sum_{i=1}^N b_i$, and let $[\mu : \nu] \in \mathbb{P}^1$ be the unique real root of the homogeneous polynomial*

$$\prod_{i=1}^N (a_i \mu + b_i \nu)^{a_i b} - \prod_{i=1}^N (a_i \mu + b_i \nu)^{b_i a} \tag{6.1}$$

*such that $a_i \mu + b_i \nu \geq 0$ for all $i \in \{1, 2, \ldots, N\}$. Let $(c_d)$ be the corresponding period sequence. Then non-zero coefficients $c_d$ satisfy*

$$\log c_d \sim Ad - \frac{\dim X}{2} \log d + B$$

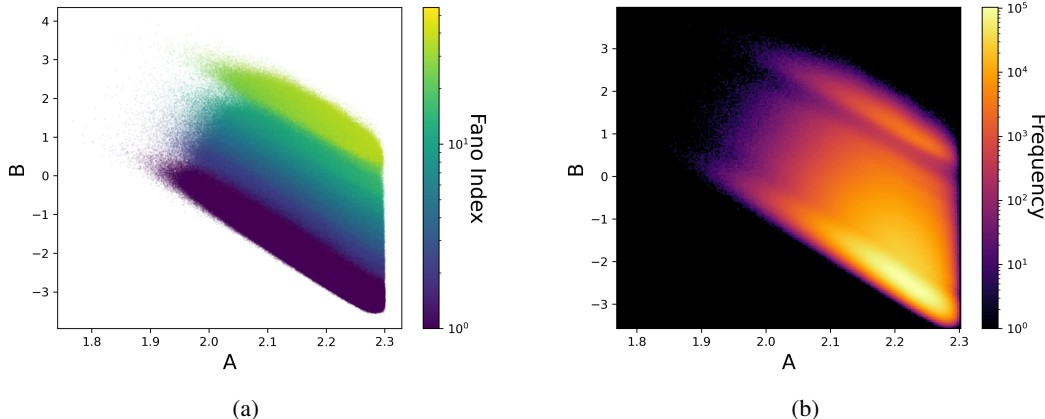

(a)                                                      (b)

Figure 3: A dataset of 100M probably-$\mathbb{Q}$-Fano toric varieties of Picard rank two and dimension eight, projected to $\mathbb{R}^2$ using the growth coefficients $A$ and $B$ from (6.2). In (a) we colour by Fano index, while in (b) we colour a heatmap according to the frequency.

*as $d \to \infty$, where*

$$A = -\sum_{i=1}^{N} p_i \log p_i$$

$$B = -\frac{\dim X}{2} \log(2\pi) - \frac{1}{2} \sum_{i=1}^{N} \log p_i - \frac{1}{2} \log \left( \sum_{i=1}^{N} \frac{(a_i b - b_i a)^2}{\ell^2 p_i} \right)$$

(6.2)

*Here $p_i = \dfrac{\mu a_i + \nu b_i}{\mu a + \nu b}$, so that $\sum_i p_i = 1$, and $\ell = \gcd\{a, b\}$ is the Fano index.*

In Figure 3 we picture our dataset of $\mathbb{Q}$-Fano varieties by using the coefficients $A$ and $B$ to project it to $\mathbb{R}^2$; for the corresponding images for terminal Fano weighted projective spaces, see [10, Figure 7a]. Note the stratification by Fano index. Although many weight matrices can give rise to the same toric variety, in our context we are using well-formed weight matrices in standard form (2.3) and so at most two weight matrices can give rise to the same toric variety. We removed any such duplicates from our dataset, so the heatmap in Figure 3b reflects genuine variation in the distribution of $\mathbb{Q}$-Fano varieties, rather than simply the many-to-one correspondence between weight matrices and toric varieties.

**Data generation**  The dataset pictured in Figure 3 was generated using an AI-assisted data generation workflow that combines algorithmic checks and our machine learning model, as follows.

- Generate a random $2 \times 10$ matrix with entries chosen uniformly from $\{0, 1, 2, 3, 4, 5, 6, 7\}$.
- Cyclically order the columns and only keep the matrix if it is in standard form, as in (2.3).
- Check conditions (1)–(4) from §3.
- Predict terminality using the neural network classifier from §4, only keeping examples that are classified as terminal and storing their probabilities.
- Set $\mu = 1$ in (6.1) and solve the univariate real polynomial in the correct domain to obtain the solution $(1, \nu)$.
- Calculate the coefficients $A$ and $B$ using the formulae in (6.2).

The final dataset is composed of 100M samples. Each of these represents a $\mathbb{Q}$-factorial toric Fano variety of dimension eight and Picard rank two that the classifier predicts is a $\mathbb{Q}$-Fano variety.

**Data analysis**  We note that the vertical boundary in Figure 3 is not a surprise. In fact, we can apply the log-sum inequality to the formula for $A$ to obtain

$$A = -\sum_{i=1}^{N} p_i \log(p_i) \leq -\left( \sum_{i=1}^{N} p_i \right) \log \left( \frac{\sum_{i=1}^{N} p_i}{N} \right) = \log(N)$$

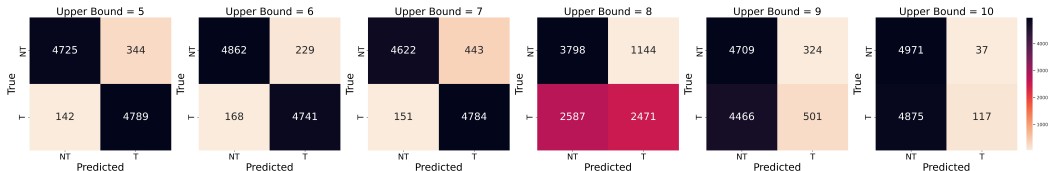

Figure 4: Confusion matrices for the neural network classifier on in-sample and out-of-sample data. In each case a balanced set of $10\,000$ random examples was tested.

In our case $N = 10$, and the vertical boundary that we see in Figure 3a is the line $x = \log(10) \sim 2.3$. We also see what looks like a linear lower bound for the cluster; a similar bound was observed, and established rigorously, for weighted projective spaces in [10].

Closer analysis (see the Supplementary Material) reveals large overlapping clusters that correspond to Fano varieties of different Fano index. Furthermore the simplest toric varieties of Picard rank two – products of projective spaces, and products of weighted projective spaces – appear to lie in specific regions of the diagram.

## 7   Limitations and future directions

The main message of this work is a new proposed AI-assisted workflow for data generation in pure mathematics. This allowed us to construct, for the first time, an approximate landscape of objects of mathematical interest ($\mathbb{Q}$-Fano varieties) which is inaccessible by traditional methods. We hope that this methodology will have broad application, especially to other large-scale classification questions in mathematics, of which there are many [1, 13, 21].

Our approach has some limitations, however, which we enumerate here. Some of these limitations suggest directions for future research. A key drawback, common to most ML models, is that our classifier performs poorly on out-of-sample data. Recall from §3 that the dataset we generated bounded the entries of the matrices by seven. For weight matrices within this range the model is extremely accurate (95%), however this accuracy drops off rapidly for weight matrices that fall outside of this range: 62% for entries bounded by eight; 52% for entries bounded by nine; and 50% for entries bounded by ten. See Figure 4 for details. Note that the network quickly degenerates to always predicting non-terminal singularities.

Furthermore the training process seems to require more data than we would like, given how computationally expensive the training data is to generate. It is possible that a more sophisticated network architecture, that is better adapted to this specific problem, might require less data to train.

Mathematically, our work here was limited to toric varieties, and furthermore only to toric varieties of Picard rank two. Finding a meaningful vectorisation of an arbitrary algebraic variety looks like an impossible task. But if one is interested in the classification of algebraic varieties up to deformation, this might be less of a problem than it first appears. Any smooth Fano variety in low dimensions is, up to deformation, either a toric variety, a toric complete intersection, or a quiver flag zero locus [9, 22]; one might hope that this also covers a substantial fraction of the $\mathbb{Q}$-Fano landscape. Each of these classes of geometry is controlled by combinatorial structures, and it is possible to imagine a generalisation of our vectorisation by weight matrices to this broader context.

Generalising to $\mathbb{Q}$-factorial Fano toric varieties in higher Picard rank will require a more sophisticated approach to equivariant machine learning. In this paper, we could rely on the fact that there is a normal form (2.3) for rank-two weight matrices that gives an almost unique representative of each $\mathrm{SL}_2(\mathbb{Z}) \times S_N$-orbit of weight matrices. For higher Picard rank $r$ we need to consider weight matrices up to the action of $G = \mathrm{SL}_r(\mathbb{Z}) \times S_N$. Here no normal form is known, so to work $G$-equivariantly we will need to augment our dataset, to fill out the different $G$-orbits, or to use invariant functions of the weights as features. The latter option, geometrically speaking, is working directly with the quotient space.

The best possible path forward would be to train an explainable model that predicted terminality from the weight data. This would allow us to extract from the machine learning not only that the problem is tractable, but also a precise mathematical conjecture for the solution. At the moment,

however, we are very far from this. The multilayer perceptron that we trained is a black-box model, and post-hoc explanatory methods such as SHAP analysis [32] yielded little insight: all features were used uniformly, as might be expected. We hope to return to this point elsewhere.

## Acknowledgments and Disclosure of Funding

TC was partially supported by ERC Consolidator Grant 682603 and EPSRC Programme Grant EP/N03189X/1. AK is supported by EPSRC Fellowship EP/N022513/1. SV is supported by the Engineering and Physical Sciences Research Council [EP/S021590/1], the EPSRC Centre for Doctoral Training in Geometry and Number Theory (The London School of Geometry and Number Theory), University College London. The authors would like to thank Hamid Abban, Alessio Corti, and Challenger Mishra for many useful conversations, and the anonymous referees for their insightful feedback and suggestions. The authors have no competing interests.

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
