# Machine learning detects terminal singularities: supplementary material

**Tom Coates**
Department of Mathematics
Imperial College London
180 Queen's Gate
London SW7 2AZ
UK
t.coates@imperial.ac.uk

**Alexander M. Kasprzyk**
School of Mathematical Sciences
University of Nottingham
Nottingham NG7 2RD
UK
a.m.kasprzyk@nottingham.ac.uk

**Sara Veneziale**[*]
Department of Mathematics
Imperial College London
180 Queen's Gate
London SW7 2AZ
UK
s.veneziale21@imperial.ac.uk

## 1    Mathematical background

**Toric varieties**    The prototypical example of a toric Fano variety is two-dimensional projective space, $\mathbb{P}^2$. As mentioned in the main text, this is defined by taking the quotient of $\mathbb{C}^3 \setminus \{\mathbf{0}\}$ by the following action of $\mathbb{C}^\times$:

$$\lambda \cdot (z_1, z_2, z_3) = (\lambda z_1, \lambda z_2, \lambda z_3)$$

The elements of $\mathbb{P}^2$ are equivalence classes that can be written as $[z_1 : z_2 : z_3]$ where at least one of the $z_i$ is non-zero. The algebraic variety $\mathbb{P}^2$ is *smooth*, since we can cover it by three open subsets that are each isomorphic to the complex plane $\mathbb{C}^2$. Namely,

$$U_1 = \{[z_1 : z_2 : z_3] \in \mathbb{P}^2 \mid z_1 \neq 0\}$$
$$U_2 = \{[z_1 : z_2 : z_3] \in \mathbb{P}^2 \mid z_2 \neq 0\}$$
$$U_3 = \{[z_1 : z_2 : z_3] \in \mathbb{P}^2 \mid z_3 \neq 0\}$$

To see that $U_1$ is isomorphic to $\mathbb{C}^2$, we note that since $z_1 \neq 0$ it can be rescaled to one. Therefore, each point in $U_1$ can be identified with a (unique) point of the form $[1 : \bar{z}_2 : \bar{z}_3]$; this gives the isomorphism to $\mathbb{C}^2$. Similar arguments show that $U_2$ and $U_3$ are each isomorphic to $\mathbb{C}^2$.

More generally, $(N - 1)$-dimensional projective space $\mathbb{P}^{N-1}$ is smooth, since it can be covered by $N$ open subsets each isomorphic to $\mathbb{C}^{N-1}$. By modifying the action of $\mathbb{C}^\times$ on $\mathbb{C}^N \setminus \{\mathbf{0}\}$ we can define more general examples of toric varieties, *weighted projective spaces*, which in general contain singular points.

For example, we can consider the action of $\mathbb{C}^\times$ on $\mathbb{C}^3 \setminus \{\mathbf{0}\}$ defined by

$$\lambda \cdot (z_1, z_2, z_3) = (\lambda z_1, \lambda z_2, \lambda^2 z_3)$$

which gives rise to the weighted projective space $\mathbb{P}(1, 1, 2)$. Here the entries of the vector $(1, 1, 2)$ are called the *weights* of the variety. In order to see that this variety is not smooth, we can consider

---

[*]Corresponding author.

37th Conference on Neural Information Processing Systems (NeurIPS 2023).

the same open sets as above,

$$U_1 = \{[z_1 : z_2 : z_3] \in \mathbb{P}^2 \mid z_1 \neq 0\}$$
$$U_2 = \{[z_1 : z_2 : z_3] \in \mathbb{P}^2 \mid z_2 \neq 0\}$$
$$U_3 = \{[z_1 : z_2 : z_3] \in \mathbb{P}^2 \mid z_3 \neq 0\}$$

As before, $U_1$ and $U_2$ are each isomorphic to $\mathbb{C}^2$. However, $U_3$ is not. In fact, since $z_3 \neq 0$ we can rescale the last entry to one, but the square in the definition of the action implies that there are two ways of doing so:

$$\pm z_3^{-1/2} \cdot (z_1, z_2, z_3) = (\pm z_3^{-1/2} z_1, \pm z_3^{-1/2} z_2, 1)$$

Therefore, $U_3 \cong \mathbb{C}^2/\mu_2$ where $\mu_2 = \{1, -1\}$ is the group of square roots of unity. Note that $\mathbb{C}^2/\mu_2$ has a singular point at the origin, which corresponds to the singular point $[0 : 0 : 1]$ in $U_3$. We say that $\mathbb{P}(1, 1, 2)$ has two smooth charts, $U_1$ and $U_2$, and one singular chart $U_3$.

This generalises to higher dimensions by considering $\mathbb{C}^\times$ acting on $\mathbb{C}^N \setminus \{\mathbf{0}\}$ by

$$\lambda \cdot (z_1, \ldots, z_N) = (\lambda^{a_1} z_1, \ldots, \lambda^{a_N} z_N)$$

for some choice of weights $(a_1, \ldots, a_N) \in \mathbb{Z}_{>0}^N$. The algebraic variety $\mathbb{P}(a_1, a_2, \ldots, a_N)$ is an $(N-1)$-dimensional $\mathbb{Q}$-factorial Fano toric variety of Picard rank one, called a *weighted projective space* [11, 14]. Setting the $a_i$ equal to 1 recovers $\mathbb{P}^{N-1}$.

For any two weighted projective spaces $X = \mathbb{P}(a_1, \ldots, a_N)$ and $Y = \mathbb{P}(b_1, \ldots, b_M)$, we can consider their product $X \times Y$. This arises as a quotient of $\mathbb{C}^{N+M}$ by an action of $\mathbb{C}^\times \times \mathbb{C}^\times$, where the first $\mathbb{C}^\times$ acts on the first $N$ co-ordinates of $\mathbb{C}^{N+M}$ and the second $\mathbb{C}^\times$ acts on the last $M$ coordinates. The two actions are specified by the weights of each weighted projective space. We can summarise this information in a *weight matrix*

$$\begin{bmatrix} a_1 & \cdots & a_N & 0 & \cdots & 0 \\ 0 & \cdots & 0 & b_1 & \cdots & b_M \end{bmatrix}$$

This type of construction can be generalised to any action of $\mathbb{C}^\times \times \mathbb{C}^\times$ on $\mathbb{C}^N$ given defined as

$$(\lambda, \mu) \cdot (z_1, \ldots, z_N) = (\lambda^{a_1} \mu^{b_1} z_1, \ldots, \lambda^{a_N} \mu^{b_N} z_N)$$

and which can be encoded in a weight matrix of the form

$$\begin{bmatrix} a_1 & \cdots & a_N \\ b_1 & \cdots & b_N \end{bmatrix}$$

Note that in the case of projective spaces and weighted projective spaces we were considering $\mathbb{C}^N \setminus \{\mathbf{0}\}$, excluding the origin because it lies in the closure of every orbit. When generalising this concept, we need to exclude more points than just the origin for the quotient to be reasonable; explicitly we consider $\mathbb{C}^N \setminus S$, where $S = S_+ \cup S_-$ for linear subspaces

$$S_+ = \{(z_1, \ldots, z_N) \mid z_i = 0 \text{ if } b_i/a_i > b/a\}$$
$$S_- = \{(z_1, \ldots, z_N) \mid z_i = 0 \text{ if } b_i/a_i < b/a\}$$

and $a = \sum_{i=1}^N a_i$, $b = \sum_{i=1}^N b_i$: see [5]. The resulting quotient $X = (\mathbb{C}^N \setminus S)/(\mathbb{C}^\times)^2$ is an $(N-2)$-dimensional toric variety. If the linear subspaces $S_+$ and $S_-$ each have dimension at least two then $X$ has Picard rank two.

**From weight matrices to fans.** In the main text, a toric variety $X$ was determined by a matrix

$$W = \begin{bmatrix} a_1 & \cdots & a_N \\ b_1 & \cdots & b_N \end{bmatrix} \tag{1.1}$$

that, as recalled above, records the weights of an action of $(\mathbb{C}^\times)^2$ on $\mathbb{C}^N$. We will now explain how to recover the fan $\Sigma(X)$ for the toric variety from this data [10, 12].

Consider the right kernel of the matrix $W$, regarded as a $\mathbb{Z}$-linear map. The kernel is a free submodule of $\mathbb{Z}^N$, of rank $N-2$, and choosing a basis for this submodule defines an $N \times (N-2)$ matrix $M$ such that $WM = 0$. The rows of $M$ define distinct primitive vectors $e_1, \ldots, e_N$ in $\mathbb{Z}^{N-2}$ such that

$$a_1 e_1 + \cdots + a_N e_N = 0$$
$$b_1 e_1 + \cdots + b_N e_N = 0$$

By construction, the vectors $e_1, \ldots, e_N$ span the kernel of $W$ over $\mathbb{Z}$.

In general the construction of a toric variety (or equivalently a fan) from a weight matrix depends also on the choice of a *stability condition*, which is an element $\omega$ of the column space of $W$. In our case, however, because $X$ is Fano there is a canonical choice for $\omega$ given by $(a, b)$, the sum of the columns of $W$. Let us denote the $i$th column of $W$ by $D_i$. We set

$$\mathcal{A}_\omega = \{I \subset \{1, 2, \ldots, N\} \mid \omega \in \angle_I\}$$

where

$$\angle_I = \left\{ \sum_{i \in I} \lambda_i D_i \ \Big| \ \lambda_i \in \mathbb{R}_{>0} \right\}$$

The fan $\Sigma(X)$ is the collection of cones in $\mathbb{R}^{N-2}$ given by

$$\{\sigma_I \mid \bar{I} \in \mathcal{A}_\omega\} \qquad\qquad \text{where } \sigma_I = \text{cone}\{e_i \mid i \in I\}$$

Here $\bar{I}$ is the complement of $I$ in $\{1, 2, \ldots, N\}$.

Recall our assumptions on the weight matrix $W$:

    0. The columns of $W$ span a strictly convex cone in $\mathbb{R}^2$.

    1. None of the columns are the zero vector.

    2. The sum of the columns is not a multiple of any of them.

    3. The subspaces $S_+$ and $S_-$, defined in the main text, are both of dimension at least two.

(We number from zero here to match the numbering of conditions in the main text.) Conditions 0 and 1 together guarantee that the fan $\Sigma(X)$ is complete; that is, its support covers $\mathbb{R}^{N-2}$. The toric variety $X$ is therefore compact. Condition 2 ensures that each top-dimensional cone in the fan has $N - 2$ rays; that is, the fan is simplicial. This implies that the toric variety $X$ is $\mathbb{Q}$-factorial. Condition 3 ensures that each of the vectors $e_1, \ldots, e_N$ generates a one-dimensional cone $\mathbb{R}_{\geq 0} e_i$ in the fan $\Sigma(X)$. Together with $\mathbb{Q}$-factoriality, this implies that the Picard rank of $X$ is two.

**Checking terminality.** Each top-dimensional cone $\sigma$ in $\Sigma(X)$ is generated over $\mathbb{R}_{\geq 0}$ by $N - 2$ of the vectors $e_1, \ldots, e_N$. These generators are contained in a unique $(N-3)$-dimensional hyperplane $H$. The cone $\sigma$ corresponds to a terminal singularity in $X$ if and only if the only lattice points in $\sigma$ that lie on or below $H$ are the generators of $\sigma$ and the origin [23]. $X$ has terminal singularities if and only if each top-dimensional cone of $\Sigma(X)$ corresponds to a terminal singularity. This justifies the assertion, given in §2 of the main text, that $X$ has terminal singularities if and only if the convex polytope $P = \text{conv}\{e_1, \ldots, e_N\}$ is mostly empty.

**A subtlety with quotient gradings.** In §1 of the main text, in the paragraph 'Why dimension eight?', we noted that the analogue of our dataset in dimension three contains 34 examples. There are 35 $\mathbb{Q}$-Fano toric varieties of Picard rank two in dimension three [15], but precisely one of these has a quotient grading and so does not fit into the framework we consider here. The exception is $X = \mathbb{P}^1 \times \mathbb{P}^2/\mu_3$, where $\mu_3$ acts via $(u, v; x, y, z) \mapsto (u, \varepsilon v; x, \varepsilon y, \varepsilon^2 z)$ and $\varepsilon$ is a primitive cube root of unity. The quotient grading arises here because the primitive generators for rays of the fan $\Sigma(X)$ fail to span the ambient lattice over $\mathbb{Z}$. If we instead regard the primitive generators as living inside the sublattice that they generate, then we recover one of the other 34 terminal examples: $\mathbb{P}^1 \times \mathbb{P}^2$. The analogue of this phenomenon happens in higher dimensions too, and so we ignore quotient gradings in our methodology.

**Significance of $\mathbb{Q}$-Fano varieties** As mentioned in §1 of the main text, $\mathbb{Q}$-Fano varieties are 'atomic pieces' from which more complicated algebraic varieties are made, and so one can think of the classification of $\mathbb{Q}$-Fano varieties as building a Periodic Table for geometry. Understanding this classification is a fundamental problem in algebraic geometry, and is the motivation behind a huge amount of research; see e.g. [4, 7, 17, 18] and the references therein.

$\mathbb{Q}$-Fano varieties also play an important role elsewhere in mathematics, for example in the study of K-stability and the existence of Kähler–Einstein metrics [2]. In theoretical physics, $\mathbb{Q}$-Fano varieties provide, through their 'anticanonical sections', the main construction of the Calabi-Yau manifolds which give geometric models of spacetime [6, 13, 22] in Type II string theory.

Moreover, terminal singularities – the focus of this paper – are the singularities that appear in the Minimal Model Program [17], and they also occur across mathematics. For example, in F-theory, terminal singularities reflect the presence of localized matter states from wrapped M2-branes which are not charged under any massless gauge potential [1]. Moreover, in the toric context, having only terminal singularities means that the corresponding polytope contains no lattice points other than the origin and the vertices. These are referred to in the combinatorics literature as one-point lattice polytopes, and are important in optimisation problems.

## 2 Further data analysis

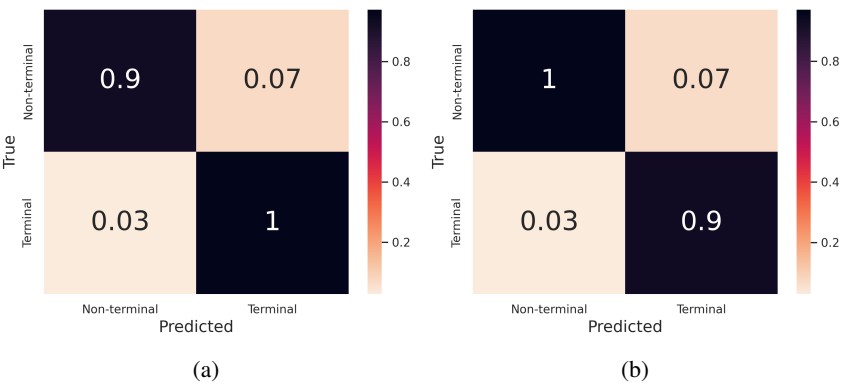

(a)          (b)

Figure 1: Confusion matrices for the classifier trained on 5M samples: (a) is normalised with respect to the true axis; (b) is normalised with respect to the predicted axis.

The neural network classifier described in §4 of the main text is remarkably accurate at determining whether a $\mathbb{Q}$-factorial Fano toric variety of Picard rank two and dimension eight is terminal or not. Confusion matrices for the classifier are presented in Figure 1. Because of this high accuracy, we were able to use this classifier to generate a dataset of 100M probably-$\mathbb{Q}$-Fano toric varieties of Picard rank two and dimension eight; see §6 in the main text. Creating this first glimpse of the $\mathbb{Q}$-Fano landscape would have been impractical using conventional methods. Based on the timing data outlined in §3 below, we estimate that generating this dataset using conventional methods would have taken 160 days on our HPC cluster, equivalent to 600 CPU *years*. In contrast, by using the neural network classifier and batch processing we were able to generate this dataset in under 120 CPU *hours*.

One striking feature of the landscape of 100M probably-$\mathbb{Q}$-Fano toric varieties, plotted in Figure 3 in the main text, is the stratification by Fano index. Recall that the Fano index of $X$ is equal to the greatest common divisor of $a$ and $b$, where $(a, b)$ is the sum of the columns of the matrix (1.1). For our dataset, the entries in the matrix (1.1) are bounded between zero and seven, and hence the range of possible Fano indices that can appear in the dataset is bounded. Figure 3 in the main text appears to show overlapping clusters of cases, with the Fano index increasing as we move from the bottom of the plot (Fano index one) to the top.

**Products of weighted projective space.** To better understand this clustering by Fano index, we consider the simplest $\mathbb{Q}$-factorial Fano toric varieties of Picard rank two: products of weighted projective spaces. Recall from §1 of this Supplementary Material that a product of weighted projective spaces $X = \mathbb{P}(a_1, \ldots, a_N)$ and $Y = \mathbb{P}(b_1, \ldots, b_M)$ is specified by a weight matrix

$$\begin{bmatrix} a_1 & \cdots & a_N & 0 & \cdots & 0 \\ 0 & \cdots & 0 & b_1 & \cdots & b_M \end{bmatrix}$$

This matrix determines a $\mathbb{Q}$-factorial Fano toric variety of Picard rank two and dimension $N + M - 2$, denoted $X \times Y$. The singular points of $X \times Y$ are determined by the singular points of $X$ and $Y$. In particular, $X \times Y$ is terminal if and only if both $X$ and $Y$ are terminal.

In general a weighted projective space $X = \mathbb{P}(a_1, a_2, \ldots, a_N)$ may have singular points; these are determined by the weights $(a_1, a_2, \ldots, a_N)$. Proposition 2.3 of [16] characterises when the singular

Table 1: The number of terminal weighted projective spaces in dimension $d$, $1 \leq d \leq 7$, with weights $a_i$ bounded by seven.

| $d$ | 1 | 2 | 3 | 4 | 5 | 6 | 7 |
|---|---|---|---|---|---|---|---|
| # | 1 | 1 | 7 | 80 | 356 | 972 | 2088 |

points of $X$ are terminal. Namely, $X$ is terminal if and only if

$$\sum_{i=1}^{N}\{ka_i/a\} \in \{2,\ldots,N-2\}$$

for each $k \in \{2,\ldots,a-2\}$. Here $a = a_1 + a_2 + \cdots + a_N$, and $\{x\}$ denotes the fractional part $x - \lfloor x \rfloor$ of a rational number $x$. This is the Picard rank one analogue to Proposition 5.3 in the main text.

We can enumerate all terminal weighted projective spaces in dimensions one to seven, with weights $1 \leq a_i \leq 7$, using the characterisation of terminal weighted projective space described above. The number in each dimension is given in Table 1. By taking products, we obtain 8792 distinct $\mathbb{Q}$-Fano toric varieties of Picard rank two in dimension eight; these examples are plotted in Figure 2. This supports our observation that the $\mathbb{Q}$-Fano varieties fall into large overlapping clusters that are determined by the Fano index. Note that the products of weighted projective space appear to fall within the upper region of each cluster.

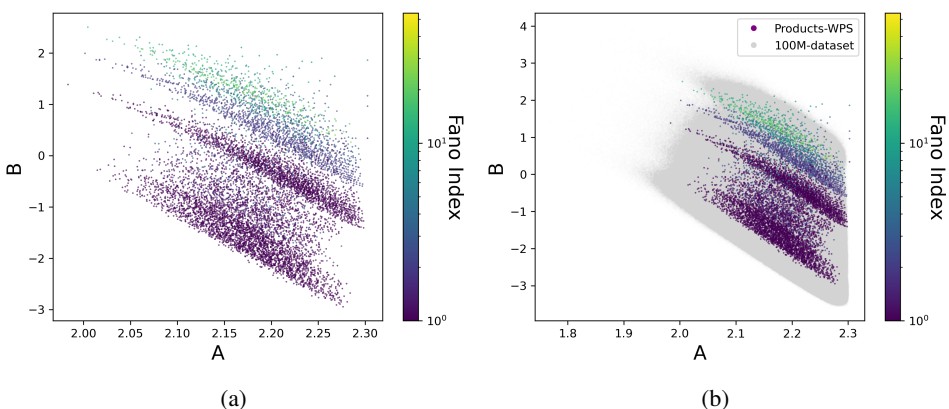

(a)         (b)

Figure 2: $\mathbb{Q}$-Fano products of weighted projective space in dimension eight, with weights bounded by seven. (a) Projection to $\mathbb{R}^2$ using the growth coefficients from (6.2) in the main text. (b) The same as (a), but plotted on top of the dataset of 100M probably-$\mathbb{Q}$-Fano toric varieties, plotted in grey.

**Smooth Fano toric varieties.** Projective space $\mathbb{P}^{N-1}$ is smooth, and so products of projective spaces are also smooth. More generally, the smooth Fano toric varieties up to dimension eight have been classified [19]. There are 62 smooth Fano toric varieties in dimension eight and of Picard rank two, all of which have weights bounded by seven when expressed in standard form (2.3) from the main text. These are plotted in Figure 3, and appear to fall in the upper extreme region within each cluster.

**A cluster of high-Fano index examples.** Figure 3 in the main text appears to show a cluster of high-Fano-index cases (at the top of the plot) standing apart from the remainder of the data. We now give an explanation for this high-Fano-index cluster. Figure 4 shows the frequency distribution of Fano indices in the dataset. The uptick in frequencies in the histogram in Figure 4 can be explained as follows. Consider how many ways we can write $N$ as a sum of ten numbers between zero and seven (inclusive, and with possible repeats). This resembles a normal distribution with $N = 35$ the most frequent case. This higher probability is due to our sampling constraints on the entries of the weight matrix: amongst those matrices that have $a = b$ we have the highest probability of selecting one that has $a = b = 35$. Therefore, we see a misleading accumulation around those Fano indices.

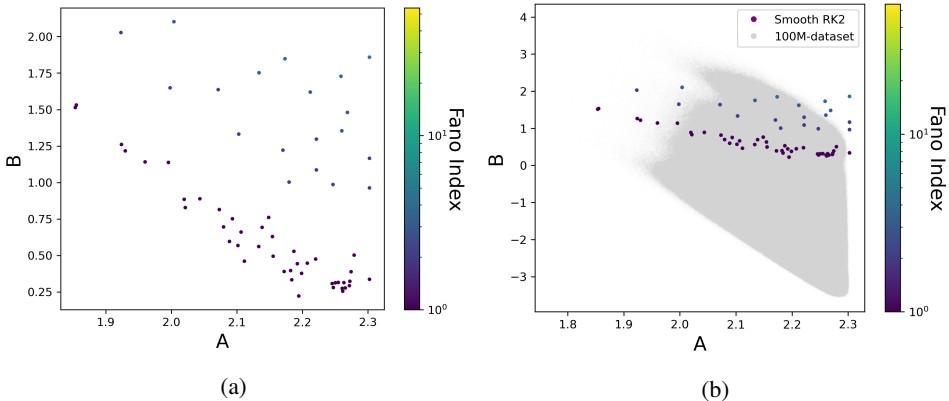

(a)                                              (b)

Figure 3: The smooth Fano toric varieties in dimension eight and of Picard rank two. (a) Projection to $\mathbb{R}^2$ using the growth coefficients from (6.2) in the main text. (b) The same as (a), but plotted on top of the dataset of 100M probably-$\mathbb{Q}$-Fano toric varieties, plotted in grey.

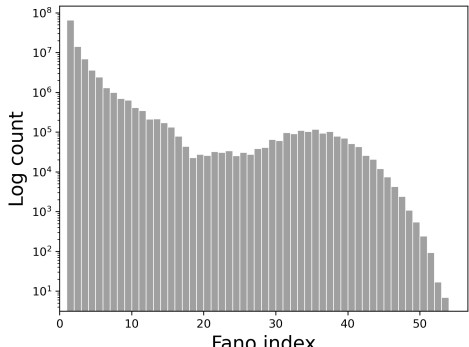

Figure 4: Distribution of the Fano index $\gcd\{a, b\}$ in the dataset of 100M probably-$\mathbb{Q}$-Fano toric varieties (note that the vertical axis scale is logged).

In Figure 5 we restrict the dataset to low Fano indices. For each Fano index in the range one through to nine, we plot the convex hull of the resulting point cloud. The overlap between these clusters is clear.

## 3   Computational resources

In this section we describe the computational resources required by different steps of our analysis. We will refer to a *desktop PC* and an *HPC cluster*. The desktop PC has an Intel Xeon 5222 quad-core processor, 64GB RAM, and an NVIDIA RTX A2000 12 GB GPU; note however that all CPU jobs on the desktop PC ran single-core. The HPC cluster has Intel Xeon E5-2650 processors with a total of 1400 cores.

**Data generation.**   The datasets `bound_7_terminal` and `bound_7_non_terminal` were generated using scripts for the computational algebra system Magma [3], running on the HPC cluster in parallel over 1400 cores for eight days, with 2GB of memory per core. Deduplication of the dataset was performed on the desktop PC and took approximately eight hours.

**Hyperparameter tuning.**   This was carried out on the desktop PC, using the GPU. Each experiment ran on average for two minutes, for a total run time of 200 minutes for 100 experiments.

**Model training.**   This was carried out using the desktop PC, using the GPU. Training on 5M balanced samples for 150 epochs took four hours.

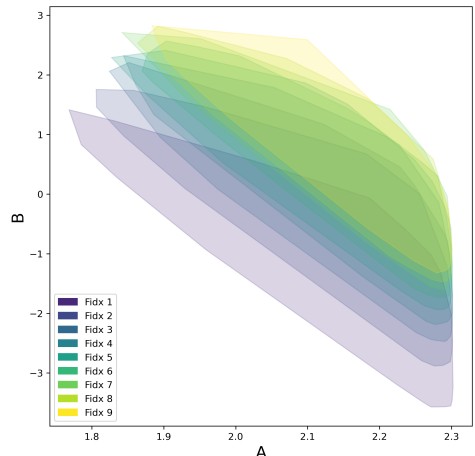

Figure 5: Convex hulls obtained from the point clouds for probably-$\mathbb{Q}$-Fano toric varieties with Fano indices between 1 and 9, obtained by projecting to $\mathbb{R}^2$ using the growth coefficients from (6.2) in the main text.

**Model evaluation.**  The model evaluation was carried out using the desktop PC, using the GPU. Evaluation took approximately ten minutes.

**Further data generation.**  The dataset `terminal_dim8_probable` was generated by running Python scripts on the HPC cluster in parallel over 120 cores for one hour, with 16GB of memory per core. Deduplication of the dataset was performed on the desktop PC and took approximately one hour.

## 4    Training for weights with a larger bound

In §7 of the main text we highlighted that the trained neural network does not perform well out of sample. Therefore, it is natural to ask whether the neural network is approximating an actual general mathematical statement, or if its performance is the result of some 'finite size effect' due to the choice of a particular weight bound (in our case seven). Our intuition here is as follows. Given that the testing and training data are free of noise (they are created through exact mathematical calculation) and the neural network classifier is so accurate, we believe that the classifier is indeed approximating a precise, general mathematical statement. However, the poor out-of-sample performance makes it unclear *what kind of mathematical statement* the network is picking up. The statement could be about weight matrices with entries of arbitrary size, or could be about weight matrices with small entries (mathematically, this would be a statement about Fano varieties with terminal singularities of bounded index). In the first case the out-of-sample performance drop-off would happen because the network is approximating the true statement in a way that does not generalise to higher weight bounds; this is a common phenomenon when developing and using neural network models. In the second case the out-of-sample performance drop-off would happen because of the underlying mathematical statement that the classifier approximates.

To probe this further, we repeated the same experiments as in the main text on a dataset of weight matrices with weights bounded by a larger constant, ten. We generated a new dataset of size 20 million, balanced between terminal and non-terminal examples, where the entries of each weight matrix are bounded by ten. The data generation steps were the same as described in §3 of the main text, except that the terminality check was now carried out using the new algorithm discussed in §5 of the main text (and proved correct in §5 of this Supplementary Material). We remark that the increased speed of the new algorithm allowed us to generate double the amount of data of the original dataset.

We used a fully-connected feed-forward neural network with the same architecture as the original neural network from the paper. This architecture is recalled in Table 2. Again, the network was trained on the features given by flattening the weight matrices, which where standardised by translating the mean to zero and rescaling the variance to one. It was trained using binary cross-entropy as loss

Table 2: Final network architecture and configuration.

| Hyperparameter | Value | Hyperparameter | Value |
|---|---|---|---|
| Layers | $(512, 768, 512)$ | Momentum | 0.99 |
| Batch size | 128 | LeakyRelu slope | 0.01 |
| Initial learning rate | 0.01 | | |

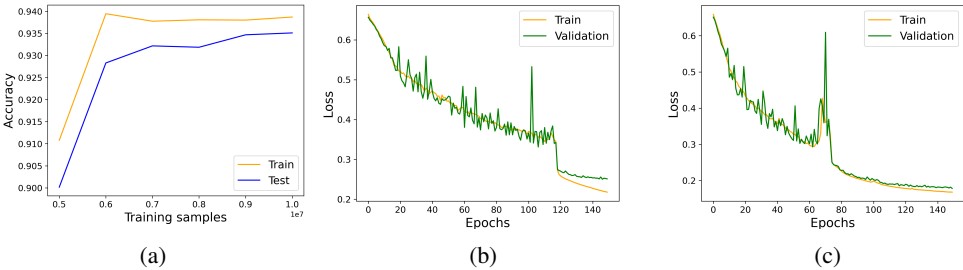

(a)    (b)    (c)

Figure 6: (a) Accuracy for different train-test splits; (b) epochs against loss for the network trained on 5M samples; (c) epochs against loss for the network trained on 10M samples.

function, stochastic mini-batch gradient descent optimiser and using early-stopping, for a maximum of 150 epochs and with learning rate reduction on plateaux.

Training on 5M samples (using 80% for training and 10% for validation) and testing on the remaining data (15M samples) produced an accuracy of 90% – see Figure 6b for the loss learning curve. This performance is worse than that achieved for the same training sample size for weight bound seven, potentially indicating that the condition approximated by the network is harder to capture. Training on a larger sample of size 10M (again using 80% for training and 10% for validation) and testing on the remaining data (10M samples) improves the accuracy to 94% – see Figure 6c for the loss learning curve. The training and validation accuracies for intermediate training sizes are shown in Figure 6a.

We were able to recover a high accuracy for this new dataset. However, this was only possible by using a larger training sample size, which hints at the increased difficulty of the task. Moreover, Figure 6a suggests that increasing the size of the training set further is unlikely to improve the accuracy. Being able to train a high-accuracy neural network for a larger weights bound supports the hypothesis that the neural network is approximating a general mathematical statement but in a way that does not generalise well to higher bounds. However, it is too early to exclude the hypothesis that the network might be capturing a mathematical statement that needs weight matrices with small entries. Similar studies with even higher bounds would add confidence here and, if the network is indeed approximating a statement about weight matrices with small weights, experiments of this type should also be able to deduce what the cut-off bound for the weights is.

**Data and code availability** The datasets underlying this work and the code used to generate them are available from Zenodo under a CC0 license [8]. Data generation and post-processing was carried out using the computational algebra system Magma V2.27-3 [3]. The machine learning model was built using PyTorch v1.13.1 [20] and scikit-learn v1.1.3 [21]. All code used and trained models are available from BitBucket under an MIT licence [9].

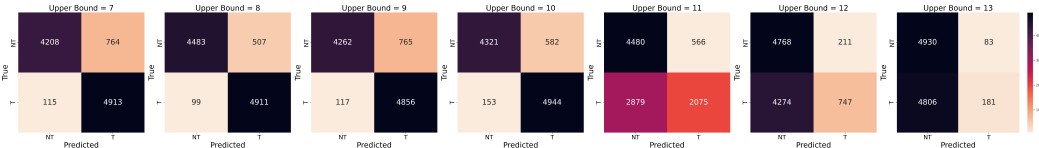

Figure 7: Confusion matrices for the neural network classifier on in-sample and out-of-sample data. In each case a balanced set of 10 000 random examples was tested.

# 5 Proof of Proposition 5.3

In this section we prove Proposition 5.3 in the main text. This is the main ingredient in the new algorithm to check terminality. Recall from the discussion above that $X$ determines a convex polytope $P$ with vertices $e_1, \ldots, e_N \in \mathbb{Z}^{N-2}$, and that

$$a_1 e_1 + \cdots + a_N e_N = 0$$
$$b_1 e_1 + \cdots + b_N e_N = 0$$

where the $a_i$ and $b_j$ are entries in the weight matrix (1.1). The same argument applied to the equivalent weight matrix

$$\begin{bmatrix} b_i/g_i & -a_i/g_i \\ A_i & B_i \end{bmatrix} \begin{bmatrix} a_1 & \cdots & a_N \\ b_1 & \cdots & b_N \end{bmatrix}$$

gives barycentric co-ordinates for the origin and $e_i$ in terms of the remaining vertices of $\Delta_i$:

$$\alpha_i^1 e_1 + \cdots + \alpha_i^{i-1} e_{i-1} + \alpha_i^{i+1} e_{i+1} + \cdots + \alpha_i^N e_N = 0$$
$$\beta_i^1 e_1 + \cdots + \beta_i^{i-1} e_{i-1} + \beta_i^{i+1} e_{i+1} + \cdots + \beta_i^N e_N = g_i e_i$$

Fix $i \in \{1, 2, \ldots, N\}$. Define $u \colon \mathbb{Q}^{N-1} \to \mathbb{Q}$ by $u(x_1, \ldots, x_{N-1}) = x_1 + \cdots + x_{N-1}$, and let $\Psi$ denote the lattice

$$\{v \in \mathcal{Z} \mid u(v) = 1\}$$

where $\mathcal{Z}$ is the span over $\mathbb{Z}$ of the standard basis $E_1, \ldots, E_{N-1}$ for $\mathbb{Q}^{N-1}$ together with

$$\frac{1}{f_i}(\alpha_1^2, \ldots, \hat{\alpha}_i^i, \ldots, \alpha_1^N) \qquad \text{and} \qquad \frac{1}{g_i}(\beta_1^2, \ldots, \hat{\beta}_i^i, \ldots, \beta_1^N)$$

Here the $\hat{\ }$ indicates that the $i$th entry in each vector is omitted. We define $\phi \colon \Psi \to \mathbb{Z}^{N-2}$ to be the $\mathbb{Z}$-linear map that sends $E_1, \ldots, E_{N-1}$ to $e_1, \ldots, \hat{e}_i, \ldots, e_N$ and

$$\phi\left(\frac{1}{f_i}(\alpha_1^2, \ldots, \hat{\alpha}_i^i, \ldots, \alpha_1^N)\right) = 0 \qquad \phi\left(\frac{1}{g_i}(\beta_1^2, \ldots, \hat{\beta}_i^i, \ldots, \beta_1^N)\right) = e_i$$

It is easy to see that $\phi$ is well-defined and bijective.

Consider the higher-dimensional parallelepiped $\Gamma$ in $\mathcal{Z}$ generated by the standard basis of $\mathbb{Z}^{N-1}$. We note that each lattice point of $\mathcal{Z}$ in $\Gamma$ can be represented as a linear combination

$$\frac{k}{f_i}(\alpha_1^2, \ldots, \hat{\alpha}_i^i, \ldots, \alpha_1^N) + \frac{l}{g_i}(\beta_1^2, \ldots, \hat{\beta}_i^i, \ldots, \beta_1^N) \tag{5.1}$$

for some $k \in \{0, 1, \ldots, f_i - 1\}$ and $l \in \{0, 1, \ldots, g_i - 1\}$; this representation is unique if and only if the vertices of $\Delta_i$ span $\mathbb{Z}^{N-2}$. Hence, $\Delta_i$ is almost empty if and only if whenever

$$\sum_{j \neq i} \left\{ k\frac{\alpha_i^j}{f_i} + l\frac{\beta_i^j}{g_i} \right\} = 1 \tag{5.2}$$

we have that the linear combination in (5.1) represents the origin. But this is the case if and only if

$$\left\{ k\frac{\alpha_i^j}{f_i} + l\frac{\beta_i^j}{g_i} \right\} = \left\{ \frac{\alpha_i^j}{\alpha_i} \right\}$$

for all $j$, since $(k, l) = (\frac{f_i}{\alpha_i}, 0)$ represents the origin by construction. Note that the sum (5.2) could include $j = i$, since that term is an integer and its fractional part will not contribute to the sum. $\quad\square$