# OpenReview forum: "Machine learning detects terminal singularities"
_NeurIPS.cc/2023/Conference — NeurIPS 2023 poster_

### Official Review · Reviewer_VZKS · 2023-06-21

**Soundness:** 3 good
**Presentation:** 4 excellent
**Contribution:** 3 good
**Rating:** 7
**Confidence:** 3

**Summary:**

The authors trained a feedforward neural network to detect $\mathbb{Q}$-Fano varieties in dimension 8, among toric varieties defined by weights in a bounded region. The resulting classifier achieves a high (95%) accuracy. The authors used this classifier to build a large dataset of (probably) $\mathbb{Q}$-Fano and visualize it. The high accuracy achieved also inspired the authors to derive a new (exact) algorithm to detect $\mathbb{Q}$-Fano varieties.

**Strengths:**

The paper is very well written, it address a problem that is interesting and probably quite important (I am not an expert in algebraic geometry though). It has a very detailed experimental part, as well as a new theoretical result (which I haven't been able to check).

**Weaknesses:**

It's not very clear whether, at the end of the day, the classifier built in this work is actually going to make an impactful contribution in the area. This weakness is outweighted by the strengths, in my opinion.

**Questions:**

I have two provocative questions.

1. Was Proposition 5.3 (or its proof) inspired in any way by the neural network? The authors seem to imply it, but that does not seem very likely. To motivate in finding this theoretical result, it seems that the important thing was to strongly believe that a simple classifier exists, not that it actually exists.

2. Regarding the poor performance on out-of-distribution examples (Figure 4): Do you think this might mean that the trained neural network is not, in fact, good enough evidence that there is a simpler algorithm to detect $\mathbb{Q}$-Fano varieties?


I also found one typo:
- Line 293: "that" -> "than"

**Limitations:**

The paper does a good job of recognizing the weaknesses in the closing section.

---

> ### Author Rebuttal · Authors · 2023-08-08
>
> Q1: *It's not very clear whether, at the end of the day, the classifier built in this work is actually going to make an impactful contribution in the area. This weakness is outweighted by the strengths, in my opinion.*
>
> A1: Understanding the classification of Q-Fano varieties is a fundamental open problem in algebraic geometry, and is the motivation behind a huge amount of important research. Techniques exist to construct certain classes (for example, the unprojection techniques in three dimensions by Reid et al in [1]) but these only scratch the surface of what is believed to exist. The ML classifier developed in this paper allows us to explore the classification at scale, giving us a "birds eye view" of the landscape that was previously impossible. Now that we know that this works (and it's important to emphasise how unexpected the success of this model was), it should be possible to construct similar models for higher Picard rank, non-toric varieties such as those described by Kalashnikov in [2], and produce a map of the Q-Fano classification. There is a precedent here: the highly influential Graded Ring Database of Brown, Reid, et al. [3]  provided the first coarse sketch of the landscape of Q-Fano 3-dimensional varieties via a (many-to-one) numerical invariant called the Hilbert series. Similarly to the ML model, their methods cannot guarantee that any given datapoint in their list is correct. Nevertheless, attempting to understand this (possible) classification  motivated a large amount of fundamental research, and led to significant breakthroughs in our understanding of Fano varieties.
>
> Q2: *Was Proposition 5.3 (or its proof) inspired in any way by the neural network? The authors seem to imply it, but that does not seem very likely. To motivate in finding this theoretical result, it seems that the important thing was to strongly believe that a simple classifier exists, not that it actually exists.*
>
> A2: The referee is correct in that we were unable to extract any precise statement from the structure of the neural network. The NN did, however, play two important roles in the development of Proposition 5.3. First, by testing ideas about when the neural network gave correct/incorrect answers, we were able to guide our understanding of the mathematical problem. For example, initially we believed that the presence of an identity block in the weight matrix would be crucial; the NN quickly disabused us of this. Without the NN guiding us here, we would have produced a considerably weaker statement. Second, as the referee suggests, simply knowing that this was possible was essential. The question of when a toric variety has terminal singularities is a well-studied and important one. And yet, beyond some elementary cases, no expert in the field suspected that this could be decided from the weight data alone. The standard (and, indeed, only) approach was to first construct the fan, and then to proceed via a cone-by-cone analysis. The striking and unexpected success of the NN provided us with the confidence to look for a result.
>
> Q3: *Regarding the poor performance on out-of-distribution examples (Figure 4): Do you think this might mean that the trained neural network is not, in fact, good enough evidence that there is a simpler algorithm to detect Q-Fano varieties?*
>
> A3: This is an interesting question which we debated at length whilst working on this paper, and is difficult to answer. It is connected with the two general questions: Can we ever expect a NN to recover a precise mathematical statement? And would a deep understanding of how a NN reaches its conclusions help constructing a mathematical result? It seems reasonable to us that the NN is approximating a more general, precise, mathematical statement; unfortunately because of the way these approximations are modelled within the structure of a NN, it doesn't give a good approximation to out-of-distribution input data. This is, of course, a common experience when developing and using ML models. Since the data is free of noise (it is purely mathematical) and the results are so accurate, we believe that the NN is detecting something which has a mathematical origin. Perhaps it has learnt something about the structure of 8-dimensional terminal singularities of low index? This is certainly possible, and one might expect the low-index cases to be "easier" to understand (although the structure in dimension 4 has only recently been understood fully; dimension 8 remains far beyond our current mathematical reach). This might explain both the efficiency of the calculation and why it fails on out-of-distribution data (which will necessarily come with higher-index singularities). We simply don't know.
>
> Q4: *Line 293: "that" -> "than"*
>
> A4: Thank you for pointing this out, this will be corrected in the revision.
>
> REFERENCES
>
> [1] Altınok S, Brown G, Reid M. Fano 3-folds, K3 surfaces and graded rings. Topology and geometry: commemorating SISTAG. 2002;314:25-53.
>
> [2] Kalashnikov E. Four-dimensional Fano quiver flag zero loci. Proceedings of the Royal Society A. 2019 May 31;475(2225):20180791.
>
> [3] Brown G, Kasprzyk A. The Graded Ring Database. Online. 2007-present.

---

> > ### Comment · Reviewer_VZKS · 2023-08-16
> > **Reply to rebuttal**
> >
> > Thanks, this all makes sense! Some of your discussion on Q2 would be a nice addition to the paper (unless it's already there and I missed it).
> >
> > I confirm my rating.

---

### Official Review · Reviewer_B8mu · 2023-06-25

**Soundness:** 4 excellent
**Presentation:** 3 good
**Contribution:** 3 good
**Rating:** 6
**Confidence:** 4

**Summary:**

The authors study properties of algebraic geometric objects, four complex dimensional Fano toric varieties,
using mathematical data science methods, i.e. statistical analysis of a large database of examples combined
with mathematical domain knowledge.  They find a fairly accurate neural network classifier to identify singular
varieties and inspired by this propose a more efficient algorithmic test.  They also study an invariant called the
quantum period.



**Strengths:**

Interesting example of mathematical data analysis for an open and actively studied problem in algebraic geometry.
The new algorithm for singularity testing is interesting.

**Weaknesses:**

The paper is very unsystematic statistically.  Since the dataset consists of mathematical objects generated by
a known sampling procedure, one could precisely characterize the ensemble being sampled in section 3 (with the
bounds on weights given there), to help interpret the results and help compare with other (future) work.  Or
(probably better) change the sampling to get a simple to describe ensemble.  This question is not even asked.
It bears on the significance of the results and the reported poor performance on out of sample test data reported in section 7.

**Questions:**

Re section 5 and algorithm 1, the motivation is the existence of an NN predictor of singularity from the weight matrix.
Did anything else from the NN results enter into finding this result?

**Limitations:**

yes

---

> ### Author Rebuttal · Authors · 2023-08-08
>
> Q1: *The authors study properties of algebraic geometric objects, four complex dimensional Fano toric varieties…*
>
> [Small correction: all our varieties are over the complex numbers, so these are 8-dimensional complex Fano toric varieties.]
>
> *The paper is very unsystematic statistically. Since the dataset consists of mathematical objects generated by a known sampling procedure, one could precisely characterize the ensemble being sampled in section 3 (with the bounds on weights given there), to help interpret the results and help compare with other (future) work.*
>
> A1: We agree with the reviewer that characterising the ensemble being sampled in Section 3 would be highly desirable. However, this is complicated by the (infinite) symmetry group involved. The space of the objects studied (weight matrices of size $r \times N$) carries the action of $GL(r, \mathbb{Z})$ on the left, together with the permutation action of the symmetric group $S_N$ on the columns. Thus the group $G = GL(r,\mathbb{Z}) \times S_N$ acts, and any two weight matrices that differ by the action of an element of $G$ represent the same toric variety. As highly non-linear objects, the $G$-orbits are hard to describe.
>
> The normal form for weight matrices that we propose and use in the paper (Equation 2.3) contains one representative of each $G$-orbit but, like all normal forms involving $GL(r, \mathbb{Z})$-equivalence, is very `non-continuous'. This is exactly what makes the Geometry of Numbers (and ultimately things like the distribution of primes, which is intimately connected with the structure of $GL(n,\mathbb{Z})$) such a challenging area of study. For example, changing a single matrix entry by plus/minus 1 will in general produce a large change in the normal form: the columns of the resulting matrix may need reordering by $S_N$, or the entire matrix may be reduced using the action of $GL(r,\mathbb{Z})$. The topology of the space given by our chosen representatives of the orbits is, therefore, counterintuitive and definitely not Euclidean.
>
> On top of this, the matrices we consider are all well-formed, which introduces further complexity. Indeed, well-formedness in rank 2 is already hard to describe from the viewpoint of geometry, and again depends on subtle considerations of lattice structure (that is, on the Geometry of Numbers). Whilst we agree with the referee that a precise characterisation of the ensemble is desirable, in practice this is far beyond our current geometric understanding of the problem.
>
> Q2: *Or (probably better) change the sampling to get a simple to describe ensemble.*
>
> A2: Again, we agree with the referee that this is a desirable step, but given the current state of understanding of the geometry, we do not believe any such rephrasing is possible.
>
> Q3: *Did anything else from the NN results enter into finding this result?*
>
> A3: As well as giving us confidence that there was a theoretical result waiting to be discovered, the NN also guided our formulation and proof of this result. For example, originally we thought that the result would only hold if the weight matrices contained an identity submatrix – the NN quickly persuaded us that this should not be the case. Without the NN model, we would have produced a considerably weaker statement. That said, the influence of the NN on the statement of the theoretical result was only indirect: we were unable to extract any precise statement from the network itself.

---

> > ### Comment · Reviewer_B8mu · 2023-08-19
> > **ack**
> >
> > Thank you for the explanations.

---

### Official Review · Reviewer_Ukzu · 2023-07-07

**Soundness:** 2 fair
**Presentation:** 2 fair
**Contribution:** 2 fair
**Rating:** 6
**Confidence:** 4

**Summary:**

This paper applies machine learning methods to predict whether an algebraic variety is Q-Fano. Q-fano varieties are Fano varieties that are Q-factorial and have terminal singularities. Checking terminally is computational challenging. To train a machine learning classifier, the authors build a balanced dataset of 10 million eight-dimensional Q-factorial Fano toric varieties of Picard rank 2. They use an MLP to classify whether a topic variety has terminal singularities (and thus a Q-fano variety) from its weight matrix. The neural network achieves an accuracy of 95%, which leads to the speculation that these exists a simple criterion to check terminal singularities directly from the weight matrix. The authors develops a new algorithm that is 15 times faster than the naive approach but still orders of magnitude slower than the ML classifier. They then conclude that the ML results suggest that a simpler criterion exists. Facilitated by the faster ML model, the authors create and visualize a dataset of 100 million examples, which provides new insights for the patterns in the classification.


**Strengths:**

- This paper provides an excellent example that shows how ML can be used to guide research in mathematics. The authors use neural networks to accelerate the classification of Fano-varieties. The computation speed by neural networks motivates new classification algorithms and suggests room for even faster ones. The trained ML model is then used as a surrogate to generate large amount of data which would be prohibitively expensive using traditional methods, and provide insights that guides theoretical understandings. While ML for science is getting popular in recent years, ML for mathematics is still a new field but has immense potential.
- The Q-factorial Fano dataset provided in this paper is a valuable assets for both math and ML communities. It will advance the understanding of Q-Fano classification and benefit future development on theorem discovery.
- The last section provides a number of interesting future directions. By relating the limitations of the current method to fundamental questions in ML (out-of-distribution generalization, data efficiency, geometric deep learning, interpretability), the authors demonstrate the importance of these topics and paves way for a new area in ML research.


**Weaknesses:**

- The mathematical objects studied in this paper is not very accessible to people without a math background. As a result, the significance of results, such as the analysis of invariant of Fano-varieties section 6, is not as strong as it could be. This is perhaps due to the large amount of prerequisite for algebraic geometry, but the authors could include more introductory materials, for example, in the appendix.
- While the AI-assisted workflow is novel, the contribution on machine learning is limited. In particular, the neural network uses multilayer perceptron, and no justification is given for the architecture choice. The theoretical result in section 5 may not be interesting to a NeurIPS audience.
- The claim that “the ML results suggests that a simpler criterion exists” for Q-Fano classification (line 82) is not well supported. The ML model does not have 100% accuracy and does not generalize when weights matrices fall outside of the range used in training. Hence the provided results does not prove that the ML model has found a simpler and correct criterion.


**Questions:**

- Would it be possible for the authors to comment more on the significance of the classification of Q-Fano varieties?
- Does detecting terminal singularities have other applications beyond a criterion for Q-Fano varieties?


**Limitations:**

The authors point out limitations in Section 7. There are no potential negative societal impacts of the work.

---

> ### Author Rebuttal · Authors · 2023-08-08
>
> Q1: *…the authors could include more introductory materials, for example, in the appendix.*
>
> A1: We agree that including additional introductory material in an appendix would be an excellent idea, and will do so in a future revision. The planned addition will cover three points. It will contextualise the relevance of algebraic geometry with respect to other fields (e.g. coding theory, algebraic vision, tensor theory, etc.); it will motivate the importance of Q-Fano varieties (see answers A4 and A5); it will give an hands on introduction to toric varieties with examples.
>
> Q2: *While the AI-assisted workflow is novel, the contribution on machine learning is limited. In particular, the neural network uses multilayer perceptron, and no justification is given for the architecture choice. The theoretical result in section 5 may not be interesting to a NeurIPS audience.*
>
> A2: We feel that including the theoretical result is important, in order to demonstrate that the AI-assisted workflow is valuable. It also highlights the difficulty of the problem that the neural network classifier is solving.
>
> About the network architecture: since the use of ML models is a novel approach in this area of mathematics, it seemed natural to try more elementary methods before delving into more advanced machine learning models. We recognise that the choice of architecture is not motivated by the problem we approach. However the result is good enough to start exploring the landscape as in Section 6, which brings new mathematical insight.
>
> In the future we would like to use network architectures that are better-suited to extracting explanations for their conclusions when applied to mathematical data. This will require improvements both in our own understanding of the techniques involved, and in the state-of-the-art for explainable models. We are hopeful that the application of AI methods to pure mathematical datasets (which are free of experimental or sampling noise) could be a fertile ground for explainable AI.
>
> Q3: *The claim that “the ML results suggests that a simpler criterion exists” for Q-Fano classification (line 82) is not well supported. The ML model does not have 100% accuracy and does not generalize when weights matrices fall outside of the range used in training. Hence the provided results does not prove that the ML model has found a simpler and correct criterion.*
>
> A3: We had meant this to read as "We believe that ... the ML results suggest that a simpler criterion exists" and will rephrase to avoid the ambiguity. We agree that our results are a long way from a proof of this statement.
>
> Q4: *Would it be possible for the authors to comment more on the significance of the classification of Q-Fano varieties?*
>
> A4: As mentioned in the Introduction, Q-Fano varieties are atomic pieces from which more complicated algebraic varieties are made, and so one can think of the classification of Q-Fano varieties as building a Periodic Table for geometry. Q-Fano varieties also play an important role elsewhere in mathematics, for example in the study of K-stability and the existence of Kähler–Einstein metrics. [1]
>
> In theoretical physics, Q-Fano varieties play an essential role in constructing models of spacetime in Type II string theory -- they provide, through their ‘anticanonical sections’, the main construction of the Calabi–Yau manifolds which give geometric models of spacetime. [2,3,4]
>
> We plan on including these remarks in the additional introduction on Fano varieties in the Supplementary materials as per A1.
>
> Q5: *Does detecting terminal singularities have other applications beyond a criterion for Q-Fano varieties?*
>
> A5: Terminal singularities are the singularities that appear in the Minimal Model Program: they are fundamental in the classification of algebraic varieties.
>
> Q-factorial terminal singularities are also important in string theory. For example, in F-theory, terminal singularities reflect the presence of localized matter states from wrapped M2-branes which are not charged under any massless gauge potential. [5]
>
> In the toric context, having only terminal singularities means that the corresponding polytope contains no lattice points other than the origin and the vertices. These are referred to in the combinatorics literature as 1-point lattice polytopes, and are important in optimisation problems.
>
> We plan on including these remarks in the additional introduction on Fano varieties in the Supplementary materials as per A1.
>
> REFERENCES
>
> [1] Berman RJ. K-polystability of Q-Fano varieties admitting Kähler-Einstein metrics. Inventiones mathematicae. 2016 Mar;203(3):973-1025.
>
> [2] Candelas P, Horowitz GT, Strominger A, Witten E. Vacuum configurations for superstrings. Nuclear Physics B. 1985 Jan 1;258:46-74.
>
> [3] Greene BR. String theory on Calabi-Yau manifolds. Fields, strings and duality. 1997;543-726.
>
> [4] Polchinski J. String theory. 2005 Jun.
>
> [5] Arras P, Grassi A, Weigand T. Terminal singularities, Milnor numbers, and matter in F-theory. Journal of Geometry and Physics. 2018 Jan 1;123:71-97.

---

> > ### Comment · Reviewer_Ukzu · 2023-08-15
> >
> > Thank you for the detailed response. I appreciate the clarifications on the significance of the Q-Fano varieties classification problem and have increased my score.

---

### Official Review · Reviewer_9Hdp · 2023-07-15

**Soundness:** 4 excellent
**Presentation:** 3 good
**Contribution:** 4 excellent
**Rating:** 7
**Confidence:** 3

**Summary:**

The paper describes an ML assisted approach to the classification of Q-Fano varieties. Checking whether the singularities of a given Q-Factorial Toric variety is very challenging by mathematically established methods, involving intensive combinatorial computations. This paper applies supervised machine learning to this problem, training a feedforward NN classifier to detect terminal varieties on a balanced dataset of 5 million 8-dimensional Q-Factorial Fano toric varieties of Picard rank 2 (there exists already a fast combinatorial formula for detecting terminality in Picard rank 1). The obtained classifier obtains a 95% test accuracy on a test set of 5 million samples from the same distribution as the training data. This is surprising because the classifier is much more efficient than known combinatorial techniques, and uses only the weight matrix corresponding to the variety. The good test performance can thus be interpreted to indicate new mathematical techniques to discover.

Taking inspiration from this, the second contribution of the paper is to develop an algorithm that detects terminality, directly from the weight matrix, for Q-Factorial Fano toric varieties of Picard rank two. This is the first algorithm that uses only the weight matrix in the case of Picard rank 2, and is measured to be ~15 times faster than existing methods. The new algorithm however, remains slower than the classifier. Additionally, weight information that is used by the new algorithm is unavailable to the NN classifier, having been destroyed by the encoding and regularization. This indicates that more mathematical discovery is possible.

Finally, the trained classifier is used as a tool to generate a dataset of "probably terminal" examples, i.e. examples that are predicted to be terminal by the classifier. This generation is much more efficient than rigorously checking terminality. A dataset of 100M probably-Q-Fano toric varieties are then investigated by performing an $\mathbb{R}^2$ projection based on the regularized quantum period. This leads to some interesting observations about the distribution of such varieties, with the authors particularly remarking on the shape of the support and the stratification by Fano index.

**Strengths:**

The paper makes a compelling and original case for the utility of machine learning in pure mathematics. It highlights that the performance of a classifier can be an indicator of the existence of undiscovered mathematical structure: the setting here is particularly interested due to the large discrepancy between the prediction and known mathematics. It also highlights the value of an efficient classifier for data generation that may allow for numerical investigations into the underlying structures. The new algorithm for detecting terminality is of possible additional interest.

The paper also does a good job of highlighting the differences between the classifier and rigorous techniques (including the new algorithm developed here). There is also a clear discussion of the current limitations of ML assisted mathematics in this setting and general directions for future improvement.

**Weaknesses:**

I do not see any glaring weaknesses. The approach as currently presented has limitations: the data requirements and poor OOD performance being the primary ones, but these are openly addressed and to be expected in early works in the field. The poor OOD performance is in my opinion the biggest obstacle to the claims of the paper, raising the troubling possibility that the better performance of the classifier comes from some 'finite size effect' due to the particular weight bound.

**Questions:**

Is it reasonable to expect that a NN classifier trained and tested on a dataset with larger weight bounds would retain the surprising performance noted in the paper? Numerically probing this may be too onerous, but if the authors could describe some intuition it would help my understanding.

**Limitations:**

Yes

---

> ### Author Rebuttal · Authors · 2023-08-08
>
> Q1: *The poor OOD performance is in my opinion the biggest obstacle to the claims of the paper, raising the troubling possibility that the better performance of the classifier comes from some 'finite size effect' due to the particular weight bound [...] Is it reasonable to expect that a NN classifier trained and tested on a dataset with larger weight bounds would retain the surprising performance noted in the paper? Numerically probing this may be too onerous, but if the authors could describe some intuition it would help my understanding.*
>
> A1: This is an important question. We have thought about it a lot during the course of this project, and do not have a clear view. We suggest that, in a future revision, we train and test a similar neural network classifier on a dataset with larger weight bounds, and report the results in the supplementary material. (It is not practical to complete this experiment by the end of the authors' response period.) Note, however, that one of the most valuable implications of our work -- the ability to sketch a part of the landscape of 8-dimensional Fano varieties that until now was completely out of reach -- holds regardless of whether the answer to the question is 'yes' or 'no'.
>
> Our intuition here is as follows. Given that the testing and training data are free of noise (they are created through exact mathematical calculation) and the neural network classifier is so accurate, we believe that the classifier is approximating a precise, general mathematical statement. But it is not clear to us whether this statement is about weight matrices with entries of arbitrary size or whether the underlying statement is something about weight matrices with small entries (mathematically, this would be a statement about Fano varieties with terminal singularities of bounded index).
>
> In the first case the OOD performance drop-off would happen because the network is approximating the true statement in a way that does not generalise to higher weight bounds; this is a common phenomenon when developing and using neural network models. In the second case the OOD performance drop-off would happen because of the underlying mathematical statement that the classifier approximates. As mentioned, one of the most valuable consequences of this approach -- our sketch of part of the landscape of higher-dimensional Fano varieties that until now was inaccessible -- remains regardless of which explanation is correct.

---

### Author Rebuttal · Authors · 2023-08-08

We thank the referees for their careful reading of the manuscript, and for their challenging and insightful questions. We have responded to each referee separately. In light of the referees' comments, we propose in a future revision to:
* add an expanded mathematical background section in the Supplementary Material. This will also include the background information included in answers A4 and A5 for reviewer Ukzu.
* add to the Supplementary Material a section discussing the results of training and testing a similar neural network classifier using weight matrices with a larger bound on the weights.
* change "that" to "than" on line 293.

---

### Decision · Program_Chairs · 2023-09-21

**Decision:**

Accept (poster)

**Comment:**

All the reviewers agreed that the the paper is well-written, the subject is quite original and important and the contributions are solid. Hence I recommend an acceptance.